# Does fungal competitive ability explain host specificity or rarity in ectomycorrhizal symbioses?

**Peter G. Kennedy**[1]*, **Joe Gagne**[1], **Eduardo Perez-Pazos**[1], **Lotus A. Lofgren**[2], **Nhu H. Nguyen**[3]

**1** Department of Plant and Microbial Biology, University of Minnesota, Minneapolis, Minnesota, United States of America, **2** Department of Microbiology and Plant Pathology, University of California, Riverside, Riverside, California, United States of America, **3** Department of Tropical Plant & Soil Sciences, University of Hawai'i, Manoa, Manoa, Honolulu, Hawai'i, United States of America

* kennedyp@umn.edu

**Data Availability Statement:** All relevant data are within the paper and its Supporting Information files.

## Abstract

Two common ecological assumptions are that host generalist and rare species are poorer competitors relative to host specialist and more abundant counterparts. While these assumptions have received considerable study in both plant and animals, how they apply to ectomycorrhizal fungi remains largely unknown. To investigate how interspecific competition may influence the anomalous host associations of the rare ectomycorrhizal generalist fungus, *Suillus subaureus*, we conducted a seedling bioassay. *Pinus strobus* seedlings were inoculated in single- or two-species treatments of three *Suillus* species: *S. subaureus*, *S. americanus*, and *S. spraguei*. After 4 and 8 months of growth, seedlings were harvested and scored for mycorrhizal colonization as well as dry biomass. At both time points, we found a clear competitive hierarchy among the three ectomycorrhizal fungal species: *S. americanus* > *S. subaureus* > *S. spraguei*, with the competitive inferior, *S. spraguei*, having significantly delayed colonization relative to *S. americanus* and *S. subaureus*. In the single-species treatments, we found no significant differences in the dry biomasses of *P. strobus* seedlings colonized by each *Suillus* species, suggesting none was a more effective plant symbiont. Taken together, these results indicate that the rarity and anomalous host associations exhibited by *S. subaureus* in natural settings are not driven by inherently poor competitive ability or host growth promotion, but that the timing of colonization is a key factor determining the outcome of ectomycorrhizal fungal competitive interactions.

## Introduction

A fundamental axiom in biology is the existence of tradeoffs, which are commonly defined as an increase in performance in one area being correlated with a decrease in performance in another area [1]. In ecology and evolution, tradeoffs generally focus on specific traits that shape the life history strategy of a given species [2]. For example, many organisms exhibit tradeoffs among growth, storage, and reproduction based on energetic constraints [3]. These tradeoffs are frequently conceptualized as bifurcations in allocation, commonly referred to as Y

**Funding:** Funding was provided by a National Science Foundation (DEB grant #1554375) to P.G. Kennedy and R. Vilgalys and the University of Minnesota Undergraduate Research Opportunity Program to J. Gagne.

**Competing interests:** The authors have declared that no competing interests exist.

models (e.g. allocation to survival versus allocation to fecundity, James [4]). While tradeoffs typically function at the level of the individual organism, their consequences can have important effects on the structure of ecological communities. Specifically, the presence of tradeoffs underpins much of the basic ecological theory explaining species coexistence [5, 6], as species generally have a particular set of traits that make them well suited to persist in certain environments but not in others.

Like many other organisms, ectomycorrhizal (ECM) fungi, which form symbiotic associations with the roots of many plants [7], have been shown to exhibit ecological tradeoffs that can be linked to their life history strategies. For example, with the rise of molecular-based analyses in fungal ecology [8, 9], it was found that many ECM fungal species that produce sporocarps abundantly aboveground were not the dominant species colonizing host root tips belowground [10]. This discordance suggested that allocation to sporocarps, which favor greater dispersal capacity and colonization of new areas, may come at the cost of proliferation of belowground mycelium [11]. Mycelial spread is essential for nutrient and water acquisition by all fungi, but for ECM fungi it is also important for colonizing additional root tips (and thereby gaining carbon from plant hosts). As such, a lower investment in mycelium would be likely to reduce root tip competitive ability for ECM fungi [12]. Subsequent empirical and theoretical studies of ECM fungal communities have provided clear support for this tradeoff between competition and colonization ability [13–15], mirroring similar tradeoffs observed in plant and microbial communities [16, 17].

A second possible competition-related trade-off for ECM fungi involves host specificity. In parasite/pathogen systems, competition for hosts is thought to select for greater parasite/pathogen specificity [18, 19], with specialists being stronger competitors than generalists on their preferred hosts (i.e. 'the jack of all trades is the master of none', 5). While specialization on a single host may counter the negative effects experienced by symbionts on hosts colonized by multiple species, other studies have shown that interspecific competition among symbionts favors the evolution of host generalism as a means of promoting local coexistence [20]. To date, there has only been one test of a putative tradeoff between competitive ability and host specificity for ECM fungi. Parlade and Alvarez [21] examined interspecific competition among four ECM fungal species, two generalists (*Laccaria bicolor* and *Pisolithus arrhizus*) and two specialists (*Rhizopogon roseolus* and *R. subareolatus*) on the common host *Pseudotsuga menziesii*. Comparing all two-species pairings, the authors found that the outcome of competition generally favored host generalists (in 3 of the 4 pairings, the generalist won). Those results suggest that competitive ability and host specificity among ECM fungi may not be tightly linked, which is consistent with many of the dominant species in ECM fungal communities being classified as host generalists [22–24].

Species abundances have also been considered in the broader ecological literature about competition-related tradeoffs. It is often assumed that rare species have low abundances because of lowered competitive abilities [25]. Despite this intuition, there are many explanations for species rarity that are not related to competitive ability [26], such as the occupation of uncommon niches [27]. A direct test of rarity and competitive ability in plants found that multiple locally rare grass species were actually strong competitors against more abundant grass species, which may facilitate their persistence despite their low abundances [28]. Similarly, Lloyd et al. [29] found that the relationship between plant range size and competitive ability was inconsistent, depending on both the lineage tested and soil fertility level. While the broader topic of rarity has received less attention in the study of ECM fungi, it has been demonstrated that less abundant species do not appear to have consistently lower nutrient acquisition potential (based on extracellular enzyme activity of ECM root tips sampled from mature trees) relative to more abundant species [30].

We recently became interested in the aforementioned tradeoffs to help understand the enigmatic life history strategy of the ECM fungus, *Suillus subaureus*. Unlike other *Suillus* species, which are largely host-specific to one of three different host genera in the family Pinaceae [31], *S. subaureus* is able to colonize both *Pinus* and *Quercus* host species [32]. However, this expansion of host range is complicated by the fact that the spores of *S. subaureus* will only germinate in the presence of Pinaceae host [32]. As such, the establishment on *Quercus* individuals requires spread of *S. subaureus* mycelium growing from colonized Pinaceae root tips. Curiously, in the three sites where we have encountered *S. subaureus* sporocarps, *Pinus* hosts are always locally absent, although they can be found as isolated individuals or in small patches less than one km away. Under those nearby *Pinus* individuals, we have encountered other *Suillus* species but never encountered *S. subaureus* despite extensive searching. Thus, it appears that the presence of *S. subaureus* on *Quercus* hosts represents a legacy effect of *Pinus* individuals that are no longer present [32]. Further, in our extensive collecting of *Suillus* throughout Minnesota, USA, we have regularly encountered many other *Suillus* species in *Pinus* forests, but only *S. subaureus* in two locations. The latter suggests that in addition to its anomalous host associations, *S. subaureus* is also a much rarer species than other *Suillus*.

To investigate how interspecific competition may contribute to the curious life history of *Suillus subaureus*, we conducted a seedling bioassay. In addition to *S. subaureus*, we collected spores of two other common *Suillus* species, *S. americanus* and *S. spraguei*, which regularly produce abundant sporocarps in *Pinus strobus* forests in eastern North America. We then inoculated *Pinus strobus* seedlings with all combinations of single- and two-species treatments of these *Suillus* species. We hypothesized that *S. subaureus* would be a competitive inferior to both *S. americanus* and *S. spraguei* based on its absence in extant *Pinus strobus* forests. From analyses of host performance in other *Suillus* bioassays [32, 33], we did not expect to observe major differences in *P. strobus* seedling biomass when colonized individually by the three different species. Finally, to better qualify the potential rarity of *S. subaureus* relative to other *Suillus* species, we compared the herbarium records and citizen science observations of these three *Suillus* species (in terms of both numbers of collections and geographic range) in Minnesota and throughout eastern North America.

## Methods

### Mushroom collection

In fall 2018, multiple sporocarps of each of the three *Suillus* species were collected at two locations in Minnesota, USA. *S. subaureus* mushrooms were collected from Interstate State Park (45.3947˚N, 92.6678˚W), while mushrooms of *S. americanus* and *S. spraguei* were collected at Cedar Creek Ecosystem Science Reserve (45.4020˚N, 93.1994˚W). Whenever possible, individual sporocarps were collected >5 m apart to try to maximize genetic diversity [34]. In the laboratory, the stipe of each sporocarp was removed and the pileus was placed onto an individual piece of aluminum foil and then covered with a glass jar. After 12 hours, the jar and pileus were both removed and the resulting spore prints were folded, placed in plastic ziptop bags, and stored at -20˚C.

### Spore inoculum

To generate spore slurries for seedling inoculations, spores of each species were washed off the aluminum foil prints using deionized water and collected into separate 50 ml tubes by species. Alcohol was used to wipe down surfaces and tools were flame-sterilized between species to eliminate cross-contamination. Under 40x magnification on a Nikon light microscope, the number of spores per slurry was counted in eight separate 10 ul aliquots using a haemocytometer, after

which the number of spores per ml was calculated. Slurries were stored at 4˚C for 24 h ahead of inoculation.

## Seedling bioassays

*Pinus strobus* seeds were obtained from Sheffield's Seed Company (Locke, NY, USA) in winter 2019. The seeds were soaked in deionized water for 24 hours and then stratified in a plastic bag at 4˚C for 30 days to facilitate germination. In April 2019, seeds were bulk planted into trays containing sterilized Sunshine grow mix soil #4 (Sun Gro, Agawam, MA, USA) and placed in a growth chamber set with a 14:10 light:dark cycle at 22˚C. For the competition bioassay, a soil mixture consisting of sand, peat moss, and forest loam soil (2:2:1 by volume) was created. The sand came from the Cedar Creek Ecosystem Science Reserve, the peat moss from a commercial source (Professional Sphagnum Peat Moss, Berger Co., Saint-Modeste, Quebec, Canada), and the forest soil from a woodland on the campus of the University of Minnesota. Both the peat moss and forest soils were sieved using a 5 mm mesh to remove sticks, roots, and other debris. The soil mixture was composited in a large heavy-duty plastic bag, shaken to homogenize, and then placed into multiple autoclave trays, spread evenly no more than 5 cm thick. To eliminate any existing fungal inoculum, soils were autoclaved at 121˚C for 90 minutes, cooled for >24 hours, and autoclaved a second time. The soil was then stored in sterile 20 L plastic buckets at 4˚C until use.

Surface-sterilized 100 mL cone-tainers (Steuwe and Sons, Tangent, OR, USA) (pots) were filled with a small amount of polyester pillow filling to cover the pot drainage holes. Each pot was then filled with 90 mL of sterilized soil. The soil was then moistened with water; some manual mixing was required due to the hydroscopic nature of the soil following autoclaving. In June 2019, one seedling was transplanted into each pot (which was harvested on the same day from the germination trays, with the root systems rinsed of any adhering media) and immediately watered to ensure root-water contact. Spores were then injected into the top layer of soil via pipetting to each treatment at a concentration of 5 x $10^5$ spores/species per ml of soil/pot. The inoculated seedlings were grown in the same growth chamber as the germination trays with the same light:dark cycle and watered 2–3 times per week throughout the duration of the experiment. The temperature in the chamber, which was set to 22˚C, fluctuated some-what during the experiment due to multiple short-term chamber malfunctions during the final month of growth, but the seedlings did not appear to suffer from this variation.

At four (September 2019) and eight (January 2020) months after spore inoculation, seed-lings in each treatment were harvested. After being removed from the pots, the root systems of each seedling were gently washed to remove all soil. The entire root system of each harvested seedling was examined under 10-20x magnification on a Nikon dissecting scope for mycor-rhizal colonization. Root tips colonized by the three *Suillus* species were distinguished from uncolonized root tips by the presence of a white mantle (S1 Fig). For the scoring of percent col-onization, all root tips less than 1 mm long were not counted. Similarly, root tips with poorly developed mantles were not counted as colonized. Any tips with coralloid clusters at their apex were counted as a single root tip. After scoring colonization, roots and shoots of each seedling were dried at 60˚C for 48 hours and then weighed.

## Molecular identification

Because the color and shape of the mycorrhizal root tips of the three *Suillus* species was very similar (S1 Fig), we used molecular techniques to identify which species were present in the two-species treatments. Ten colonized tips per seedling in each two-species treatment were randomly selected and placed into 10 μl of tissue Extraction Solution (Sigma Aldrich,

St. Louis, MO, USA) (if less than 10 tips were colonized, then all tips were removed). The tubes containing the tips and Extraction Solution were placed onto a thermocycler and incubated at 65˚C for 10 minutes and 95˚C for 10 minutes. Afterwards, 30 µl of Neutralization Solution were added to each tube and all extractions were stored at 4˚C ahead of PCR. The Internal Transcribed Spaced (ITS) region of the rRNA gene was amplified using the fungal-specific primer pair ITS1F [35] and ITS4 [36]. 15 µl PCR reactions were conducted using 7.5 µl of REDE PCR mix, 0.75 µl of each primer (at [10 µM]), 1 µl of DNA template, and 5 µl of sterile water. PCR cycling conditions were as follows: 95˚C for 1 minute, then repeating 95˚C 30 seconds, 55˚C for 20 seconds, and 72˚C for 50 seconds, 34 times, then 72˚C for 5 minutes and cooled to 12˚C. Success of PCR was confirmed by running 5 µl of reaction product on a 1% agarose gel. Restriction Fragment Length Polymorphism (RFLP) reactions were then performed on the products from successful PCRs using the restriction enzyme *Alu*l and Cutsmart buffer (New England Biolabs, Waltham, MA, USA). Gel electrophoresis was run on the products using 2%/1% (regular/low melt temperature) agarose gel. RFLP band patterns were compared to patterns from known samples of each species (S2 Fig).

## Statistical analysis

All statistical analyses were conducted in R (v3.6.2 [37]). The percent mycorrhizal colonization of each *Suillus* species in the two-species treatments was determined by taking the total percent mycorrhizal colonization and dividing it by the ratio of the counts for each species on the 10 tips analyzed (i.e. if 9 of 10 tips belonged to *S. subaureus* and 1 belonged to *S. spraguei*, 90% of the total mycorrhizal colonization would be assigned to *S. subaureus* and 10% of the total mycorrhizal colonization to *S. spraguei*). For each species, the percent mycorrhizal colonization was compared using Type-II sum of squares two-way analyses of variance (ANOVAs) in the 'car' package (https://cran.r-project.org/web/packages/car/car.pdf), with harvest date and treatment (single- versus two-species) as the predictor variables. At each harvest date, total seedling biomass was compared using a one-way ANOVA, with inoculation treatment as the predictor variable. Before running all ANOVAs, variance and normality assumptions were tested and found to be met, except for the percent mycorrhizal colonization by *S. spraguei*, which was dominated by zero values. Differences among means were assessed using a post-hoc Tukey HSD Test run in the 'emmeans' package (https://cran.r-project.org/web/packages/emmeans/emmeans.pdf).

## Herbarium collections

On March 15, 2020, the Mycology Collections Portal (www.mycoportal.org) was queried for records of *S. subaureus*, *S. americanus*, and *S. spraguei*. Nguyen et al. [31] determined that *S. spraguei* is the correct name for North American collections labeled as *S. pictus*, so here records of *S. spraguei* and *S. pictus* were combined. We made two separate queries: 1) the number of collections and their geographic locations for collections held in the Bell Museum of Natural History, which primarily covers the region of Minnesota, USA and 2) the number of collections and their geographic locations for collections currently in any of the following herbaria, which cover the entire geographic range of *Pinus strobus*: State University of New York College at Cortland (CORT), Eastern Illinois University (EIU), Field Museum of Natural History (F), University of Illinois Herbarium (ILL), University of Illinois, Illinois Natural History Survey Fungarium (ILLS), Indiana University (IND), Iowa State University, Ada Hayden Herbarium (ISC), University of Michigan Herbarium (MICH), Michigan State University Herbarium (MSC), NAMP—New York Mycological Society: Macrofungi of New York City, New York (NAMP-NYMS), New York Botanical Garden (NY), New York State Museum (NYS), State

University of New York, SUNY College of Environmental Science and Forestry Herbarium (SYRF), Royal Ontario Museum Fungarium (TRTC), University of Wisconsin-Stevens Point Herbarium (UWSP), University of Manitoba (WIN), University of Wisconsin-Madison Herbarium (WIS). On the same date, we also searched for observations of all three *Suillus* species on iNaturalist (inaturalist.org). In both datasets (i.e. collections and observations), we limited our final records to those within the native range of *P. strobus*.

## Results

After 4 months, the number of seedlings colonized varied depending on treatment (Table 1). In the single-species treatments, 100% of the seedlings inoculated with *S. americanus* were colonized, 85% of the seedlings inoculated with *S. subaureus* were colonized, and none of the seedlings inoculated with *S. spraguei* were colonized. In the two-species treatments, all seedlings except for one seedling in the *S. subaureus*/*S. spraguei* treatment were colonized. In terms of mycorrhizal colonization, all three species showed no significant differences among the single- and two-species treatments (Fig 1A–1C), despite trends indicating that *S. subaureus* was the competitive inferior to *S. americanus* and that *S. spraguei* was competitively inferior to both other *Suillus* species due to its lack of colonization in all treatments.

After 8 months, the colonization frequency across treatments remained similar for *S. americanus* and *S. subaureus*, but differed for *S. spraguei* (Table 1). As opposed to zero colonization after 4 months, *S. spraguei* colonized the majority of the seedlings in the single-species treatment (5 of 7), although at a consistently low percent mycorrhizal colonization (mean = 4%, Fig 1F). *S. spraguei* also colonized a single tip on one of the seedlings in the *S. subaureus*/*S. spraguei* treatment, but was not detected on any of the other seedlings in that treatment and was completely absent from the *S. americanus*/*S. spraguei* treatment. The amount of mycorrhizal colonization of *S. americanus* remained similar across treatments (indicating no significant negative effect of competition) (Fig 1D), while the percent mycorrhizal colonization of *S. subaureus* was significantly lower in the *S. americanus*/*S. subaureus* treatment than the single-species treatment and the *S. subaureus*/*S. spraguei* treatments (Fig 1E). Additionally, the percent

**Table 1. Frequency of ectomycorrhizal fungal colonization and total biomass of *Pinus strobus* seedlings in the competition bioassay.**

| Growth Period | Competition Treatment | Species Inoculation | Seedling Replicates | Seedlings Colonized | Seedling Biomass (g) |
|---|---|---|---|---|---|
| 4 month | N/A | No inoculation | 7 | 0 | 0.17 (0.02) a |
| 4 month | Single-species | *S. americanus* | 7 | 7 | 0.23 (0.03) a |
| 4 month | Single-species | *S. subaureus* | 7 | 6 | 0.23 (0.02) a |
| 4 month | Single-species | *S. spraguei* | 7 | 0 | 0.15 (0.02) a |
| 4 month | Two-species | *S. americanus—S. subaureus* | 7 | 7 | 0.19 (0.03) a |
| 4 month | Two-species | *S. subaureus—S. spraguei* | 7 | 6 | 0.16 (0.01) a |
| 4 month | Two-species | *S. americanus—S. spraguei* | 7 | 7 | 0.14 (0.02) a |
| 8 month | N/A | No inoculation | 7 | 0 | 0.38 (0.06) AB |
| 8 month | Single-species | *S. americanus* | 7 | 7 | 0.49 (0.06) A |
| 8 month | Single-species | *S. subaureus* | 7 | 6 | 0.41 (0.04) AB |
| 8 month | Single-species | *S. spraguei* | 7 | 5 | 0.39 (0.05) AB |
| 8 month | Two-species | *S. americanus—S. subaureus* | 7 | 7 | 0.31 (0.03) AB |
| 8 month | Two-species | *S. subaureus—S. spraguei* | 7 | 4 | 0.29 (0.04) AB |
| 8 month | Two-species | *S. americanus—S. spraguei* | 7 | 7 | 0.21 (0.03) B |

Different letters signify significant differences as determined by Tukey HSD tests (letter sizes correspond to separate ANOVAs).

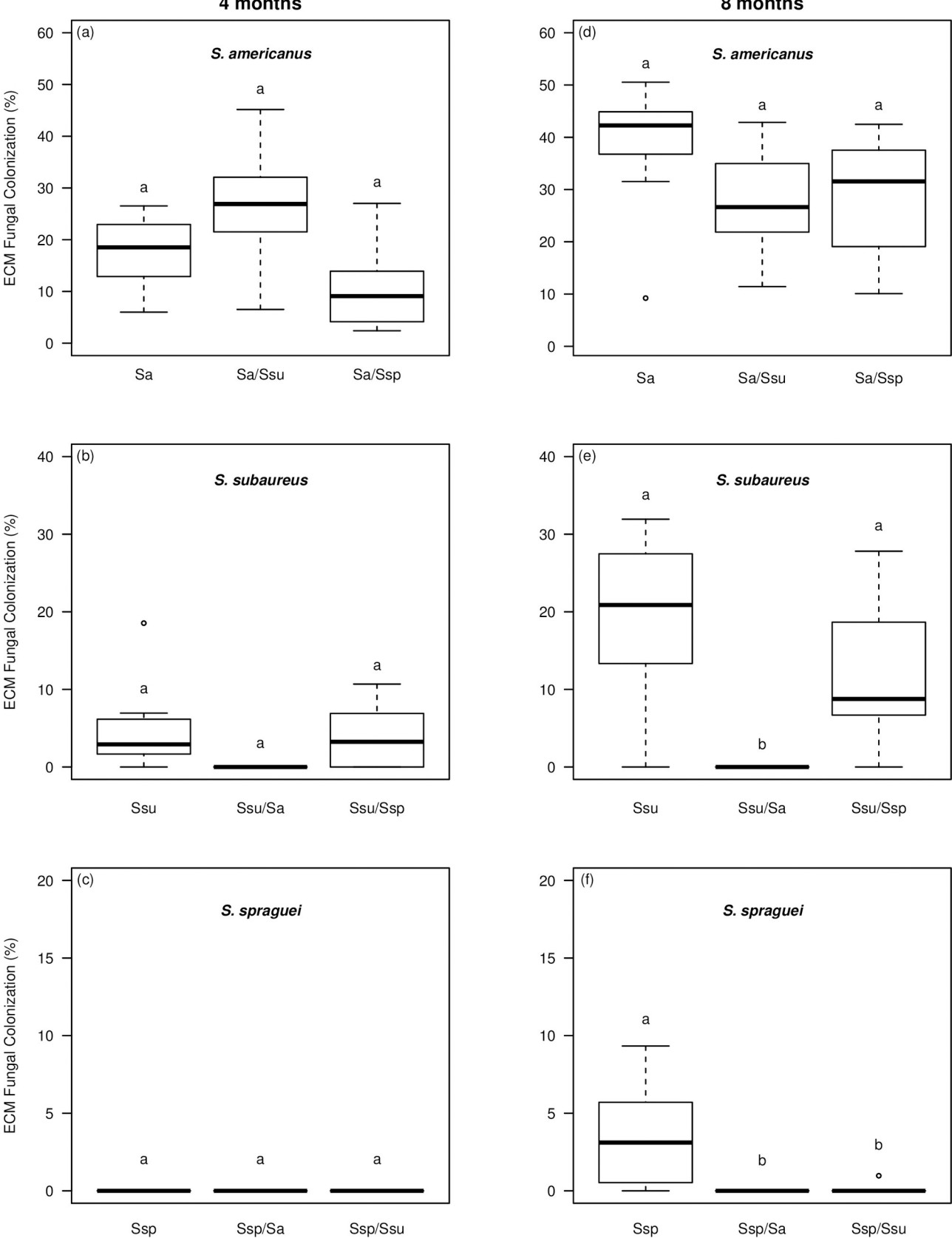

**Fig 1. Percent ectomycorrhizal fungal colonization by *Suillus americanus* (a,d), *S. subaureus* (b,e), and *S. spraguei* (c,f) on *Pinus strobus* seedlings after 4 and 8 months of growth.** Boxes represent the 2nd and 3rd interquartile ranges; the horizontal lines in the boxes represent the median; the upper and lower bars outside the boxes represent the 1st and 4th quartiles, respectively. Letters indicate significant statistical differences (p < 0.05) between means based on Tukey HSD tests. N = 7 for each treatment at both time points.

mycorrhizal colonization of *S. spraguei* was significantly higher in the single-species treatment than in either of the two-species treatments (Fig 1F).

The total biomass of seedlings after 4 months was not significantly different across treatments, including the non-inoculated control seedlings (Table 1). After 8 months, seedlings in the *S. americanus* single-species treatment had significantly higher total biomass than the seedlings in the *S. americanus*/*S. spraguei* treatment, with the mean total biomass of seedlings in all other treatments being intermediate.

There was a total of 85 records for the three *Suillus* species in the Bell Museum herbarium (Minnesota, USA), 324 records in the combined herbaria of eastern North America, and 1689 records in the iNaturalist database. While the geographical range of all three species was generally similar in both east-west and north-south extent (Fig 2), *S. subaureus* was the least abundant species in all three queries. Specifically, *S. subaureus* represented only ~20% (60/324) of the records in the combined herbaria in eastern North America, ~6% (5/85) of records in the Bell Museum herbarium, and ~1% (22/1789) of the iNaturalist observations. *S. americanus* was more than twice as common in both the Bell and other herbaria collections as *S. spraguei* (Bell: *S. americanus* = 56, *S. spraguei* = 24; Other herbaria: *S. americanus* = 179, *S. spraguei* = 80),

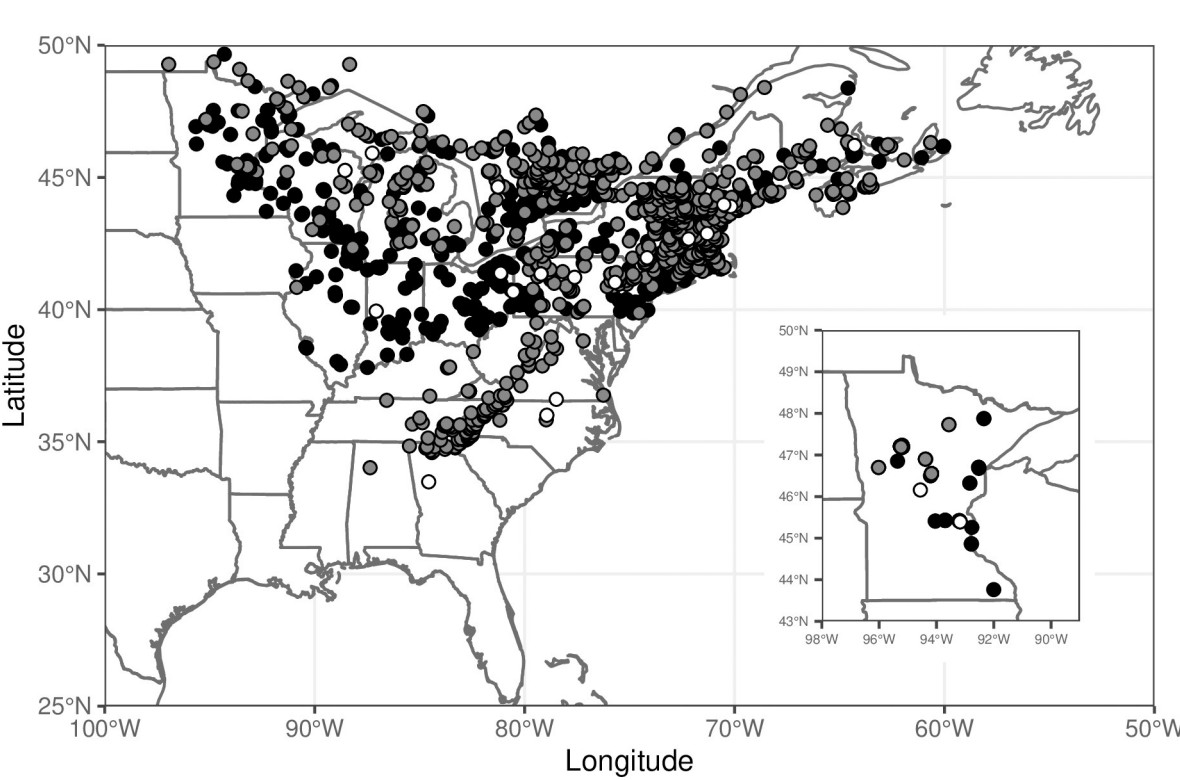

**Fig 2. Distribution of the three *Suillus* species based on observations in the iNaturalist database (main map) and collections in Minnesota from the Bell Museum of Natural History Herbarium (inset).**

while the two species had relatively similar abundances in the iNaturalist observations (*S. americanus* = 772, *S. spraguei* = 895).

## Discussion

We found a clear competitive hierarchy among the three ECM fungal species studied, with *S. americanus* being competitively dominant, *S. subaureus* competitively intermediate, and *S. spraguei* competitively inferior. This ordering is not consistent with our hypothesis that *S. subaureus* would be the competitive inferior due to its absence from extant *P. strobus* forests. While our results only pertain to a limited set of congeneric ECM fungal species, they do suggest that the local absence of *S. subaureus* in *P. strobus* forests is not likely due to inherently low competitive ability. This finding is consistent with other studies of rare species, which indicate that species abundance and competitive ability are not necessarily tightly coupled [28, 29]. In our study system, the unusual capacity of *S. subaureus* to colonize *Quercus* individuals creates a competitive refuge, at least from other Pinaceae-specific ECM fungal species, which are among the dominant colonists of young pine seedlings in Minnesota and elsewhere [38]. We suggest these two factors, i.e. sufficient competitive ability against some other ECM fungal species and the unique ability to colonize novel hosts, combine to support the overall persistence of *S. subaureus* mixed conifer-angiosperms forests in spite of small population sizes.

The competitive dynamics observed among these *Suillus* species reveal important parallels with other studies of ECM fungal competition. For example, it appears that the competitively dominant species, here *S. americanus*, possesses spores that germinate both rapidly and consistently. This was most apparent at the first harvest (i.e. four months), where *S. americanus* colonized every seedling onto which it was inoculated (including in the two-species treatments), while *S. subaureus* colonized many but not all of the seedlings onto which it was inoculated, and *S. spraguei*, which colonized none of the seedlings onto which it was inoculated. Because early colonization of a single root tip facilitates additional root tip colonization via mycelium on the same seedling [39], this early and consistent colonization likely explains the higher levels of percent mycorrhizal colonization of *S. americanus* across treatments. The results from the second harvest confirm that all three species had viable spores, as *S. spraguei* was found in the majority of the single-species seedling replicates and on a single root tip of one *S. subaureus*/*S. spraguei* seedling. However, the delay in colonization by *S. spraguei* that resulted in notably lower percent mycorrhizal colonization, particularly relative to the other two species, is likely a significant factor explaining why it was outcompeted in all two-species treatments. The slower spore germination by *S. spraguei* may also be linked to its preference for older *P. strobus* forests, in contrast to *S. americanus* which is more common in younger *P. strobus* forests and where rapid spore germination is particularly important for initial establishment (N. Nguyen, pers. obs.). The presence of preemptive colonization driving competitive outcomes, a.k.a. priority effects, appears to be a common pattern observed in ECM fungal competition, both in lab and field settings (see Kennedy [12] for a review). Yet other studies have found that competitive outcomes can be reversed depending on order or arrival [40] or when based on mycelial rather than spore colonization [13]. We imagine that these latter possibilities in combination with the highly patchy nature of ECM fungal assemblages [41, 42] allow for sufficient opportunities for *S. spraguei* to colonize *P. strobus*, thereby preventing competitive exclusion in natural settings.

A second possible explanation for the absence of *S. subaureus* in extant *P. strobus* forests is that it is not a beneficial symbiont. Recent work on plant discrimination in ECM fungal symbioses suggests that plants can reward portions of the root system that provide more nitrogen and that there is some amount of pre-colonization chemical screening that controls ECM

fungal colonization [33]. However, in the case of *S. subaureus*, it does not appear that there is active host inhibition, as it colonized many of the seedlings just as quickly as *S. americanus*. Similarly, the lack of differences in seedling dry biomass across treatments is also consistent with *S. subaureus* being a functionally equivalent symbiont to the other two *Suillus* species. The soil mixture used is not particularly nutrient rich, but it is possible that the experimental conditions used masked potential growth effects across treatments, especially since none of the inoculated seedlings were significantly larger than the non-mycorrhizal control seedlings (Table 1). We did, however, find that ECM fungal competition lowered seedling growth, which is similar to previous results in other studies of ECM symbioses [14, 43]. Moving forward, more detailed studies using isotopic tracers to track carbon and nitrogen exchange ([e.g. 44]) will be key to determining whether *S. subaureus* has different physiological interactions with its host relative to other species. Based on our results here and those presented in Lofgren et al. [32], however, we suspect that symbiont benefit is not a primary driver of the anomalous host associations of *S. subaureus*.

Both the herbarium records and citizen science observations of these three *Suillus* species correspond well with our own field observations. Specifically, we have found *S. subaureus* fruiting only 3 times in 5 years of regular mushroom collecting in the midwestern USA, while we have found both *S. americanus* and *S. spraguei* fruiting every year during the same time period in a wide variety of locations in the midwestern USA. Members of the genus *Suillus* produce relatively large sporocarps are generally considered prolific fruiters [34], so we doubt that the patterns observed for *S. subarueus* are due to cryptic missed collections. Similarly, the distinctive coloring of *S. subauerus*, which has a more orange pore surface and different pileus characteristics than other co-occurring *Suillus* species (S3 Fig), makes it unlikely that the differences in collections and observations reflect misidentification. Instead, we believe that *S. subaureus* is truly a rare *Suillus* species. While fungal conservation efforts lag behind those of plants and animals, there is growing global interest in documenting fungal species abundances for assigning protected status [45]. Although neither host of *S. subaureus* appears to be in need of conservation, given the limited abundance of *S. subaureus* and its particularly unique life history, it may be a good candidate for being designated at greater conservation status.

## Conclusions

This study provides new insight into the curious ecology of the rare ectomycorrhizal fungus, *S. subaureus*. Our results suggest that two apparent hypotheses are unlikely to explain its anomalous host associations: 1) a weak competitive ability forcing *S. subaureus* to take refuge on alternative hosts and 2) a poor symbiont that is actively discriminated against by its *Pinus* host. Future competitive tests against other species common in ECM fungal spore banks [38] and mature *Pinus* forests [46] will help in determining the extent to which the results observed here hold against more distantly related species. Additionally, comparisons of the genome content of *S. subaureus* [47] as well as its gene expression during symbiotic establishment [48] relative to other *Suillus* and ECM fungal species will help shed light on its unique capacity to associate with both *Pinus* and *Quercus* hosts.

## Supporting information

**S1 Fig. *Pinus strobus* root tips at 20x magnification.** (A) Uncolonized. (B) Colonized by *Suillus subaureus*. (C) Colonized by *Suillus americanus*.
(DOCX)

**S2 Fig. Restriction length fragment polymorphism (RFLP) digest patterns for three *Suillus* species digested using *Alu*1.**
(DOCX)

**S3 Fig. Mushrooms of the three *Suillus* species.** (A) *Suillus subaureus*. (B) *Suillus americanus*. (C) *Suillus spraguei*.
(DOCX)

**S1 Table.**
(CSV)

**S1 Data.**
(CSV)

**S2 Data.**
(CSV)

**S3 Data.**
(CSV)

## Acknowledgments

We thank A. Certano and F. Maillard for assistance with the bioassay set-up and maintenance.

## Author Contributions

**Conceptualization:** Peter G. Kennedy, Lotus A. Lofgren, Nhu H. Nguyen.

**Data curation:** Joe Gagne, Eduardo Perez-Pazos.

**Formal analysis:** Peter G. Kennedy, Joe Gagne.

**Funding acquisition:** Peter G. Kennedy, Nhu H. Nguyen.

**Methodology:** Joe Gagne, Eduardo Perez-Pazos.

**Project administration:** Peter G. Kennedy.

**Resources:** Lotus A. Lofgren, Nhu H. Nguyen.

**Supervision:** Peter G. Kennedy.

**Validation:** Nhu H. Nguyen.

**Visualization:** Eduardo Perez-Pazos.

**Writing – original draft:** Peter G. Kennedy.

**Writing – review & editing:** Peter G. Kennedy, Joe Gagne, Eduardo Perez-Pazos, Lotus A. Lofgren, Nhu H. Nguyen.

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
