## [Decision Letter · Decision Letter 0]

6 Jul 2020

Does fungal competitive ability explain host specificity or rarity in ectomycorrhizal symbioses?

PONE-D-20-14490

Dear Dr. Kennedy,

We’re pleased to inform you that your manuscript has been judged scientifically suitable for publication and will be formally accepted for publication once it meets all outstanding technical requirements.

Kind regards,

Cheng Gao

Academic Editor

PLOS ONE

Additional Editor Comments (optional):

Reviewers' comments:

Reviewer's Responses to Questions

**Comments to the Author**

1. Is the manuscript technically sound, and do the data support the conclusions?

Reviewer #1: Yes

Reviewer #2: Yes

2. Has the statistical analysis been performed appropriately and rigorously? 

Reviewer #1: Yes

Reviewer #2: Yes

3. Have the authors made all data underlying the findings in their manuscript fully available?

Reviewer #1: Yes

Reviewer #2: Yes

4. Is the manuscript presented in an intelligible fashion and written in standard English?

Reviewer #1: Yes

Reviewer #2: Yes

5. Review Comments to the Author

Reviewer #1: This is a good paper on the relationship between ectomycorrhizal (EM) fungal competitive ability and fungal specificity or rarity, by using two common EM fungal species (Suillus americanus and Suillus spraguei) and one rare species (Suillus subaureus) in a seedling bioaasay. The results indicated that fungal identity has no effect on dry biomasses of seedlings, and the timing of fungal colonization is important for the results of fungal interspecific competition. Meanwhile, the The study shed a new light on the tradeoff between fungal specificity and its competitive ability, and I believe many fungal biologists will find these results interesting and the paper deserves to be published. In general, the manuscript is well-written and clear, and the methods, statistical analysis and results seem very robust. However, I have a few suggestions of my own. They are all fairly minor but I feel that they should be addressed prior to publication.

Line 158: add the reference for the “>5 m apart to try to maximize genetic diversity”

Line 277: “In” should be corrected as “in”

Tables 1: I am not familiar with the format of Tables in this journal, but I suggest you use three-line form to make it looking better.

Reviewer #2: Your article is very interesting, and I also learned a lot from it. Here are some minor suggestions, hoping to help you improve your article.

1. Your article is innovative, however the references you cited are a little out of date. Please add more recent references (especially in the introduction part).

2. The whole article is very organized and logical, which makes it easy for readers to follow. However, some parts may be described a little too much. For example, in page 4 lines 80-85, you described enzymatic activities produced by ECM fungi, but it doesn't seem to have much to do with your article.

All in all, your article is interesting and of great quality, it is a pleasure to participate to its review.

6. PLOS authors have the option to publish the peer review history of their article (what does this mean?). If published, this will include your full peer review and any attached files.

Reviewer #1: Yes: Yonglong Wang

Reviewer #2: No

---

## [Editor Report · Acceptance letter]

23 Jul 2020

PONE-D-20-14490 

Does fungal competitive ability explain host specificity or rarity in ectomycorrhizal symbioses? 

Dear Dr. Kennedy:

I'm pleased to inform you that your manuscript has been deemed suitable for publication in PLOS ONE. Congratulations! Your manuscript is now with our production department. 

Kind regards, 

on behalf of

Dr. Cheng Gao 

Academic Editor

PLOS ONE